# Acute Effects of Ocrelizumab Infusion in Multiple Sclerosis Patients

**DOI:** 10.3390/ijms232213759

**Published:** 2022-11-09

**Authors:** Katja Akgün, Johanna Behrens, Dirk Schriefer, Tjalf Ziemssen

**Affiliations:** Center of Clinical Neuroscience, Department of Neurology, University Hospital Carl Gustav Carus, Technical University Dresden, 01309 Dresden, Germany

**Keywords:** ocrelizumab, infusion-related reactions, multiple sclerosis, adverse events

## Abstract

B cell-depleting therapies such as ocrelizumab (OCR) are highly effective in people with multiple sclerosis (MS). Especially at treatment start and initial infusion, infusion-related reactions (IRR) are a common adverse event. The relevance of acute changes of cell-depleting therapies on peripheral immune compartments and routine lab testing is important for clinical practice. We systematically analyzed routine blood parameters, detailed blood immunophenotyping and serum cytokine profiles in 45 MS patients starting on OCR. Blood samples were collected before and after corticosteroid premedication and directly after each OCR infusion of the first three ocrelizumab infusions. Blood B cells were rapidly depleted and accompanied only by a mild cytokine release at the first OCR infusion. Cytokine release was not significantly detectable from a third application in line with decreasing IRRs. B cell depletion was accompanied by short-lived changes in other immune cell populations in number, activation and cytokine secretion after each OCR infusion. Standard lab parameters did not show any clinically relevant changes. Our data demonstrate only mild changes during the first OCR infusion, which are not present any more during long-term treatment.

## 1. Introduction

Ocrelizumab (OCR) is a humanized monoclonal anti-CD20 antibody approved for relapsing and primary progressive forms of multiple sclerosis (RMS resp. PPMS). The clinical trials OPERA I and OPERA II demonstrated that OCR is highly effective in RMS patients to prevent clinical and subclinical disease activity and disease progression compared to interferon-beta treatment [1,2]. The ORATORIO trial confirmed that significantly fewer PPMS patients on OCR experienced disability progression and walking impairment in contrast to the placebo group [3,4]. The mechanism of action is mediated by the selective depletion of CD20+ cells and by focusing on B cell subtypes especially in the peripheral compartment [5,6]. By binding to CD20, which is mostly expressed in middle-aged B cells, OCR causes lysis and depletion by antibody-dependent cell-mediated cytotoxicity, antibody-dependent cellular phagocytosis as well as complement-dependent cytotoxicity and apoptosis [5,6]. During OCR therapy, B cell reconstitution by B cell precursors and pre-existing humoral immunity are preserved due to the fact that B cell precursor cells and plasma cells do not express CD20 [5,6]. A small subset of pro-inflammatory T cells that express CD20 are also effectively decreased and supposed to contribute to efficacy in addition to B cell lysis [7,8]. Current data propose no relevant impact on innate and adaptive immunity, especially in the peripheral immune compartment beyond B cells [7,9,10]. However, an increased infectious risk on OCR infusion has been described, and clinical trial data confirm higher incidence of appearance of respiratory and urinary tract infections as well as herpes-virus related infections during OCR therapy, but none directly linked to infusion [1,3,4,11]. Little is known about peripheral immune cell subsets immediately after OCR infusion. Infusion-related reactions (IRR) are a common phenomenon in depleting strategies using monoclonal antibodies and are mostly linked to cytokine release syndrome. These acute effects and changes to the peripheral immune compartment can severely interfere with routine laboratory evaluations, as seen with alemtuzumab in MS [12,13]. Within the clinical trials, IRR were the most frequently documented adverse events in OCR-treated patients, particularly after the first dose and subsequently decreasing with following applications [1,3,4]. Premedication including corticosteroids and antihistamines are recommended to avoid IRR in OCR-treated patients and to increase tolerability [14]. Since the approval of OCR, and in the real world cohort, further questions arise: Are OCR-mediated effects on cytokine release and immune cell subsets linked to clinical signs of IRR and in need of prophylactic premedication? To what extent do acute OCR-mediated changes interfere with the peripheral immune cell compartment and routine lab testing and do these changes impact the interpretation of routine lab testing for patients management? These aspects are of so much importance in clinical use and everyday clinical practice as we used to treat an older cohort with relevant comorbidities and other immunosuppressive pretreatment constellations compared to patients in clinical trials.

In this study, we systematically evaluated acute versus long-term effects on routine blood parameters, detailed blood immunophenotyping and serum cytokine profiles in 45 MS patients starting on OCR for the first infusions up to 12 months. We question how acute versus follow-up changes may impact clinical presentation and management of IRR and if a period of relevant infectious risk differs during treatment time. 

## 2. Results

### 2.1. Stable Vital Signs and Mild Adverse Events

There were no relevant changes in blood pressure and heart rate during the infusion procedure at all investigated time points (Table 1). Body temperature was stable without significant change at all time points, and the highest documented body temperature was 37.7 °C (Table 1). Infusion-associated adverse events were mild (grade 1): 34% of patients during the first infusion, 22.5% during the second infusion and 10% during the third and fourth infusion presented IRR. Headache (20%), dizziness (7.5%), sore throat (5%), nausea (5%) and exanthema (5%) were the most common IRR during the first OCR infusion. In 17.6% of IRR cases, infusion was occasionally interrupted or infusion flow rate reduced. All patients completed planned infusions. During the 12-month observation period, only mild to moderate (grade 1 and 2) adverse events appeared in 12.5% of patients presenting respiratory or urinary tract infections not linked to infusion application. 

### 2.2. Acute Effects on Complete Blood Count 

Directly after each OCR infusion, leukocyte count increased in all treated patients but declined to normal until follow up (Figure 1A). This effect was most pronounced after the first OCR infusion and could be primarily attributed to increased neutrophil counts (r = 0.995, *p* < 0.05), as the number of eosinophils and basophils dropped (Figure 1B–D). Absolute monocytes and activated monocytes decreased after each infusion (Figure 1E,F). The baseline lymphocyte count was on a lower level and 43% of patients presented lymphocytes below the reference range of 1.5 GPT/L. After each OCR infusion, lymphocytes significantly dropped (Figure 1G). Directly after the first OCR infusion, about 90% of patients presented a lymphocyte count < 0.5 GPT/L, and 15% of patients even < 0.2 GPT/L. Although lymphocyte count recovered before the next infusion, individual levels of lymphocyte counts were decreased by about 18.6% at months 3, compared to the baseline. No changes were detectable for hemoglobin, hematocrit, erythrocyte and platelet count (data not shown). Changes of complete blood counts during OCR infusion were not statistically associated with the appearance of IRR in our cohort in a multivariate evaluation adjusted for age and sex.

### 2.3. Clinical Chemistry

OCR infusions did not have an acute or prolonged impact on routine laboratory parameters (Appendix A). Liver enzymes such as gamma-GT, ALAT, ASAT, pancreatic-amylase and kidney function parameters such as creatinine, glomerular filtration rate, as well as sodium and potassium levels (data not shown) remained stable within physiological ranges during and after infusion. There was no clinically significant increase of acute phase proteins CRP and PCT after corticosteroid or OCR infusion. Single patients presented with a transient increase of CRP < 40 mg/mL, but this was not associated with IRR signs. LDH as marker of cell death did not relevantly change during the treatment period. Complement activation was evaluated by complement C3 and C4 serum levels, which were stable during the whole infusion cycle (data not shown).

### 2.4. Effects on Selective Immune Cell Subsets

There were no significant effects directly after methylprednisolone (MP) infusion on the CD19^+^ B cell, CD3^+^ T cell, CD4^+^ T cell, CD8^+^ T cell, NK cell or NKT cell count (Figure 2A–F). OCR infusion led to a dramatic and rapid drop in absolute and relative CD19^+^ B lymphocytes without significant repopulation before the next infusion (Figure 2A). B cells were not detectable directly after initial OCR infusion in 33.4% of patients, in 5.1% of patients at month three and 10.2% of patients at month nine. In all patients, positive B cells could be detected at month six and twelve before the next infusion but were still relevantly decreased ≤ 0.005 GPt/L in 71.4%and 72.7% of patients at months six and twelve, respectively. Repopulation of B cell count by >0.010 GPt/L after initial depletion was found in 14.3% of patients at months six and 9.1% of patients at months twelve. Within the blood B cell compartment, there was no change in relative count of naïve and memory B cells (Figure 2G,H). Regulatory B cells and plasmablasts were relatively and absolutely decreased after OCR (Figure 2I,J). Directly after the OCR start, CD20+ B cell subsets (B naive and memory cells, plasmablasts, Breg cells) significantly decreased (data not shown) in a prolonged manner. Annexin and viability dye (VD) staining of CD19+ B cells presented dying and dead cells early after the first OCR infusion and even at follow-up visits (Figure 2K,L). 

Interestingly, also absolute CD3^+^, CD4^+^ and CD8^+^ T lymphocytes significantly decreased directly after each OCR infusion, but had quickly recovered two weeks after the three-month follow up (Figure 2B–D). This acute CD4+ T cell decrease was correlated with the intensity of CD19 B cell depletion (r = 0.676, *p* < 0.05). After each OCR infusion, the relative count of CD3+ cells decreased, but the proportion of CD4+ and CD8 T cells including naïve (CD45RA+) and memory (CD45RO+) subsets were stable (Figure 2M). A subset of CD20+ T cells (CD3+, CD4+ and CD8+) decreased to a low but remained at a stable level after the first OCR infusion (Figure 2N). The proportion of IFN-gamma-releasing Th1 cells was significantly reduced directly after the first OCR infusion, whereas Th2, Th17 and Treg subsets were not affected by OCR infusions (Figure 2O). Evaluation of apoptosis and necrosis markers could not confirm relevant T cell death at all evaluated time points (data not shown). Absolute NK cells and NKT cells dropped significantly after the first OCR infusion, but recovered before the second infusion and remained stable in the reference range (Figure 2E,F). Furthermore, absolute NK cell decrease was correlated with CD19 B cell depletion (r = 0.593, *p* < 0.05). Relative distribution of NK cell subsets (CD56^low^CD16^−^; CD56^low^CD16^bright^; CD56^bright^CD16^−^) did not differ during the evaluation period (data not shown). This effect on B cell, T cell and NK cell subsets during OCR infusion were consistent in all patients. The appearance of IRR was not statistically associated with the intensity or time point of the acute changes on the presented peripheral immune cell subsets. 

Although the absolute count of monocytes decreased after each OCR infusion (Figure 1), the relative count of monocyte subsets (classical, intermediate, non-classical monocytes) did not significantly change (data not shown). The absolute and relative count of various DC populations such as slanDCs, myeloid and plasmacytoid DCs remained stable.

### 2.5. Cytokine Levels and Cell Activation Level upon Stimulation

MP alone had no impact on serum cytokine levels, but the first OCR infusion lead to a significant but transient increase in IL-6, IFN-gamma and TNF-alpha in all investigated patients (Table 2). Cytokine levels decreased after the initial dose of OCR and were not significantly elevated during follow-up infusions. IL-10 peaked after the first and also after follow-up infusions. TNF-alpha and IL-6 increase correlated with each other (r = 0.68, *p* < 0.05). However, the appearance of IRR was not statistically associated with the intensity or time point of the serum cytokine release and though could not predict clinical appearance of IRRs. The TNF-alpha release was negatively correlated with the depletion of CD19 B cells after the initial OCR dose (r = −0.576, *p* < 0.05). There was no correlation between the OCR-induced cytokine release and parameters of standard lab testing including CRP and PCT or other immune cell subsets. 

In addition, we evaluated the serum cytokine levels of three patients without MP pretreatment during the third OCR infusion. In these patients, IL-1beta, IL-6, IL-10, IFN-gamma and TNF-alpha peaked again at a maximum one fold compared to levels before infusion. Patients were free of any relevant clinical IRRs. 

Prior to the first infusion, cytostim stimulation led to an increase of IL-2, IL-10, IL17A, IL-22, IFN-gamma and TNF-alpha in supernatants of stimulated cells and was not impaired by MP pretreatment (Figure 3A–E). OCR infusion led to acute and prolonged inhibition in IFN-gamma, IL-2 and TNF-alpha release in stimulated cells. Stimulation with LPS induced an increase in IL-1beta, IL-6, IL-10, IL-12 and TNF-alpha (Figure 3D–H). This LPS-induced cytokine release was inhibited for IL-beta, IL-10 and IL-12 after OCR application (Figure 3D,F,G). No cytokine expression was detectable in unstimulated cells (data not shown).

## 3. Discussion

OCR is highly effective against multiple sclerosis because of the selective depletion of CD20+ cells [1,2,3,15]. Many depleting monoclonal antibody therapies are associated with IRRs at different severities that occur within the initial 24 h and are linked to acute cytokine release after initiated cell lysis [12,13,16], although most monoclonal antibody treatment schemes suggest premedication as antihistamines and corticosteroids to avoid IRRs. In clinical trials, the occurrence of IRRs were documented in about 35% of patients during OCR infusion over the entire study period [1]. These data are in line with our cohort initially presenting IRRs in 34% of patients with a continuous decrease during subsequent infusions. The incidence of IRRs during B cell depleting therapies is much lower compared to combined B/T cell depleting strategies as seen during alemtuzumab treatment with up to 90% of IRRs in treated patients [17,18]. Cytokine release and lymphocyte activation could account for such IRRs [12,17]. Our data confirm moderate release of inflammatory cytokines after initial OCR infusion, which presents much lower compared to cytokine levels after CD52-targeted depletion, although B cell lysis is fast and nearly complete after the first OCR infusion [13]. However, levels of serum cytokines as well as acute changes in peripheral lymphocyte subsets were not linked to the appearance of clinically evident IRRs. The increase of cytokines during follow-up infusions were only rarely detected, which is in line with the significantly fewer reported IRRs after the third OCR dose. Acute management of occurred IRRs includes slowing the infusion rate and additional application of corticosteroids, or at least stopping infusion [16]. The infusion rate was slowed in 17% of patients that presented acute IRRs, and OCR dosing could be completed in all patients at each visit. Certainly, cytokine release profiles did not differ in these patients. It is supposable that sustained low B cell count through frequent OCR application lowers the risk of IRRs because of the reduced amount of cell lysis during follow-up infusions. Recently, a shorter OCR infusion time after initial dosing was approved for patients that tolerate infusions well and did not present IRRs during previous infusions [19]. In our study, we included three patients without corticosteroid pretreatment prior to the third infusion. None presented with IRRs, and there was only a slight, but not clinically significant increase in IL-1beta, IL-6, IL-10, IFN-gamma and TNF-alpha after OCR infusion. 

Our data confirm only minimal cytokine release after the initial two OCR doses and demonstrate excellent tolerability even without corticosteroid pre-treatment. We can suggest not to use a corticosteroid pretreatment in patients with relevant comorbidities, including diabetes mellitus or cardiovascular diseases.

The OCR-mediated depletion of CD20 expressing cells is rapid and substantial within minutes after the initial infusion, as seen in all of our analysed patients. Over 30% of patients presented no detectable B cells after initial OCR dose. Comparable with other studies even at the time of subsequent doses, the B cell number was significantly decreased ≤ 0.005 GPT/L in >70% of patients and B cells were repopulated by >0.010 GPt/L in only about 10% of patients before the next infusion [7,10,20,21]. This effect is consistent in all CD20+ B cell subsets, including naïve B cells, memory B cells, regulatory B cells and plasmablasts. B cell-specific apoptosis and cell deaths were found to be the highest after the initial dose, but also at the three- and six-month follow ups, suggesting sustained B cell specific antibody efficacy even after infusion, which conforms with the known half-life of 26 days [1,14]. The distribution of naïve vs. memory B cell subsets were unchanged at our observed time points. These results are in contrast to other reports discussing a decrease in naïve subtypes during OCR and may be explained by differences in gating strategies and timing of sample collection [7]. Further reports demonstrated changes of cytokine release and B cell activation after the OCR start [7,22]. These data support the idea of an immune reconstitution-like profile after B cell depletion with OCR. Previous reports discussed that T cell activation is altered during anti-CD20 therapy [22]. With the exception of CD20+ T cells, no changes in T cell numbers during OCR treatment could be proven after focusing on six monthly evaluation intervals [7,10,23]. CD20+ T cells are supposed to be highly active and pro-inflammatory cell subsets contributing to MS pathology [24]. In line with recent data, we detected a direct reduction since the first OCR infusion in CD20+ CD4+ and CD8+ T cell subsets [7,23]. Interestingly, the absolute count of further T cell subsets and IFN-gamma and IL-2 cytokine release was not only significant, but transiently decreased after each OCR infusion. Evaluation of apoptosis markers could not prove relevant T cell death during infusion. As previously shown, aberrant B cell activation and pro-inflammatory cytokine profiles can mediate pro-inflammatory T cell responses in MS that can be affected by B cell depleting strategies [22]. Although depletion of CD20+ T cells is assumed to be a consequence of the OCR-induced cell lysis, we propose short-term impaired T cell activation of further subsets caused by the acute effects on B cells after each OCR therapy. The distribution of naïve, memory and effector T cells was stable throughout the whole observation period. However, functional properties including pro-inflammatory cytokine release of T cells after activation were decreased over the whole observation period compared to baseline. These durable effects on T cells may contribute to OCR efficacy in addition to the direct B cell-depleting effect. 

Directly after each OCR infusion, the absolute leukocyte neutrophil count increased. This effect is most likely attributed to the corticosteroid pretreatment as previously discussed [13]. As patients undergoing cell-depleting therapies are at higher risk for infectious diseases, the evaluation of laboratory parameters are important in addition to clinical monitoring of infectious symptoms. In addition to complete blood count, acute phase proteins such as CRP and PCT are useful markers of viral and bacterial infections. However, cytokine storms in cell-depleting therapies and comorbidities including liver failure have been reported to lead to misinterpretation of their test results [13,25,26]. Our data present that CRP and PCT can serve as valid markers, even during the post-infusion period. 

Innate immune cell subsets such as monocytes, eosinophils, basophils and NK cells decreased and the activation/maturation profile changed after each OCR infusion. There are single reports about serious infections including opportunistic infections in acute B cell-depleted patients [20,27]. Our data present a transient decrease in immune competence directly after each OCR infusion, potentially compromising adaptive and innate immunity. This effect is only short lived and comes to baseline quite soon. Infections early after OCR application have to be seriously examined as patients with comorbidities or disabilities must be frequently evaluated, and interventions such as surgeries should be preplanned and the combination with other immunosuppressants such as repeated intrathecal or i.v. corticosteroid use should be critically discussed. 

Taken together in our real world study, we present rapid and profound cellular depletion of B cell subsets accompanied by a moderate cytokine release only after the first OCR infusion. Our data demonstrate that the first dose infusion of OCR causes IRR in a subset of patients but does not recur with subsequent treatments. In addition to acute B cell depletion, T cell subsets and innate immune cells are affected for a short period after each OCR infusion, which could account for a higher risk of infections after infusion.

## 4. Materials and Methods

### 4.1. Patients and Study Approval

In our study, we included 45 patients diagnosed with RMS or PPMS treated with OCR after critical review of clinical and MRI data and extensive discussion of available treatment options (Table 3). The experiments were approved by the institutional review board of the University Hospital of Dresden. Patients gave their written informed consent.

### 4.2. OCR Infusion Protocol and Blood Sampling

The OCR infusion protocol used in our MS center was based on the standardized infusion protocol defined by the European Medicines Agency (EMA) [14]. Before the treatment start, contraindications were checked and active infections were excluded. An initial dose of 600 mg OCR was given on two days intravenously (i.v.), at 300 mg each at intervals of 14 days. The following doses of 600 mg each were applied every six months over the course of four hours. On each infusion day, patients were pre-treated with 100 mg i.v. methylprednisolone (MP) followed by three hours of OCR infusion. Patients received prophylactic anti-histaminergic treatment with 5 mg of oral desloratadine twice daily starting the day before infusion for three days. Monitoring of vital signs including blood pressure, heart rate and body temperature were performed hourly until one hour after the end of the infusion. Blood samples were taken at baseline prior to MP, after MP and directly after OCR administration on each of the infusion days as well as during the three-month follow-up visit. In addition, three patients were evaluated without MP pretreatment since the third OCR infusion because of known diabetes mellitus and a significant increase in blood glucose levels during corticosteroid treatment. 

### 4.3. Routine Blood Analysis

Routine blood parameters were assessed at the Institute of Clinical Chemistry and Laboratory Medicine, University Hospital in Dresden, Germany. The institute complies with standards required by DIN-EN-ISO-15189 for medical laboratories. Routine blood testing included complete blood cell count, liver enzymes, pancreatic-amylase, creatinine, sodium and potassium, and acute-phase proteins including C-reactive protein (CRP), procalcitonine (PCT), lactate dehydrogenase (LDH) and complement components C3 and C4. 

### 4.4. Immune Cell Phenotyping by Fluorescence-Activated Cell Sorting (FACS)

After blood collection, peripheral blood mononuclear cells (PBMC) were prepared by Ficoll-Hypaque (Biochrom, Berlin, Germany) density centrifugation. Subpopulations of T-cells, B-cells, natural killer (NK) cells and antigen-presenting cells (APC) were characterized by surface staining with fluorescence labelled anti-CD3, anti-CD4, anti-CD5, anti-CD8, anit-CD10, anti-CD16, anti-CD14, anti-CD19, anti-CD20, anti-CD25, anti-CD27, anti-CD38, anti-CD45RA, anti-CD45RO, anti-CD127, anti-CD138, anti-HLADR, (BD Biosciences, Heidelberg, Germany) or anti-BDCA1, anti-BDCA2, anti-slan (Miltenyi Biotec, Bergisch-Gladbach, Germany) according to the manufacturer’s instructions. Viability dey (VD, eBioscience, SanDiego, USA) and Annexin V (Miltenyi Biotech) staining was used to evaluate apoptotic and necrotic cells. For additional characterization of intracellular markers, PBMCs were suspended in a culture medium consisting of RPMI 1640 (Biochrom), 5% human AB serum (CC pro, Neustadt, Germany), 2 mM L-glutamine, 100 U/mL penicillin and 100 µg/mL streptomycin (Biochrom). To evaluate cytokine release and T cell polarisation, PBMC were stimulated with 10 ng/mL phorbol myristate acetate (PMA, Sigma-Aldrich, St. Louis, MI, USA) and 1 µg/mL ionomycin (Sigma-Aldrich) in the presence of 0.2 µM Monensin (Biomol, Hamburg, Germany) for 6 h prior to analysis. Before characterization of intracellular markers, cells were fixed with freshly prepared fixation concentrate, and permeabilized with wash-permeabilisation concentrate (Fixation/Permeabilisation Buffer Set, eBioscience). Subsequently, cells were stained using fluorescence labelled anti-interferon (IFN)-gamma, anti-interleukin (IL)-4, anti-IL-17A and anti-tumor necrosis factor (TNF) alpha (BioLegend, London, UK) or anti-FoxP3 antibody (Miltenyi Biotec). Negative controls included directly labelled or unlabelled isotype-matched irrelevant antibodies (BD Biosciences). Cells were evaluated on LSR-Fortessa (BD Biosciences). Gating strategy for cells is presented in Appendix A.

### 4.5. Cytokine Assay

Serum samples were taken and directly frozen (−80 °C) after collection. Afterwards, serum was analysed to obtain concentrations of interleukin (IL)-1beta, IL-2, IL-4, IL-5, IL-6, IL-9, IL-10, IL-12, IL-13, IL-17A, IL-22, IFN-gamma and TNF-alpha using a commercial multiplexed fluorescent bead-based immunoassay (eBioscience, Frankfurt, Germany) according to the manufacturer´s instructions. Supernatants were collected and the concentrations of cytokines were determined using the commercial multiplexed fluorescent bead-based immunoassay as described above.

To investigate the potential of PBMCs to release cytokines upon stimulation ex vivo after OCR infusions, freshly prepared PBMC from a subgroup of 13 patients were suspended in a culture medium consisting of RPMI 1640 (Biochrom, Cambridge, UK), 5% human AB-serum (CC pro, Neustadt, Germany), 2 mM L-glutamine, 100 U/mL penicillin and 100 µg/mL streptomycin (Biochrom), plated on round-bottomed 96-well plates at 2 × 10^5^ cells/well and maintained for 18 h in a cell culture. Cytostim (Miltenyi Biotec) was added for the last four hours to stimulate cytokine release in T lymphocytes; lipopolysaccharide (LPS) (1 µg/mL, Sigma Aldrich) was included for the complete 18 h to stimulate cytokine release in B lymphocytes and innate immune cells. Unstimulated cells served as controls. Supernatants were collected and the concentrations of cytokines were determined using the commercial multiplexed fluorescent bead-based immunoassay as described above.

### 4.6. Statistical Analysis

To characterize the overall study population, socio-demographic (age, sex) and the disease-specific data (MS type, disease duration, EDSS score at treatment initiation) were collected. Associated quantitative variables were presented as mean and standard deviation (SD) and/or 95% confidence interval (CI), whereas categorical variables were expressed as absolute (n) and relative frequencies (%). 

Normality of the data was visually inspected using quantile–quantile plots and histograms in combination with Shapiro–Wilk tests. For the outcomes of main interest (dependent variables), generalized linear mixed models (GLMMs) were applied to longitudinally analyze the course over each of three time points (before infusions started (A), after MP (B) and after OCR (C); time as independent variable). In addition, GLMM models were extended by adding not only time (A, B, C), but also IRR (yes, no) and the interaction between time and IRR into the model. In the case of normally distributed outcomes, a GLMM with normal distribution and an identity link function was used. For right- skewed data, a log transformation was either used in advance to achieve normality, or a GLMM with gamma distribution and log link function was implemented, depending on the nature of the data. Left-skewed data were reflected in the right-skewed distribution (mirror transformation) and then analyzed in the same way as the right-skewed data. For pairwise comparisons (contrast tests), Bonferroni correction was utilized to account for multiplicity. Significant results were those with (adjusted) significance levels of *p* < 0.05. In the figures, *p*-values of contrast tests and global F-tests were reported as significant with * *p* < 0.05, ** *p* < 0.01 and *** *p* < 0.001, respectively. To examine the interdependence of the main outcome variables both cross-sectionally and over time (A to C), pairwise Spearman correlations were calculated, thus taking into account the skewed distribution of some outcome variables. Statistical analyses were performed using IBM SPSS version 25.0 (IBM Corporation, Armonk, NY, USA) and GraphPad Prism 7.0 (GraphPad Software, Inc), the latter used to generate all figures. Plausibility checks were performed prior to the analyses to identify any potential individual measurement errors in the laboratory. No statistical imputation procedures for missing values were applied.

## Figures and Tables

**Figure 1 ijms-23-13759-f001:**
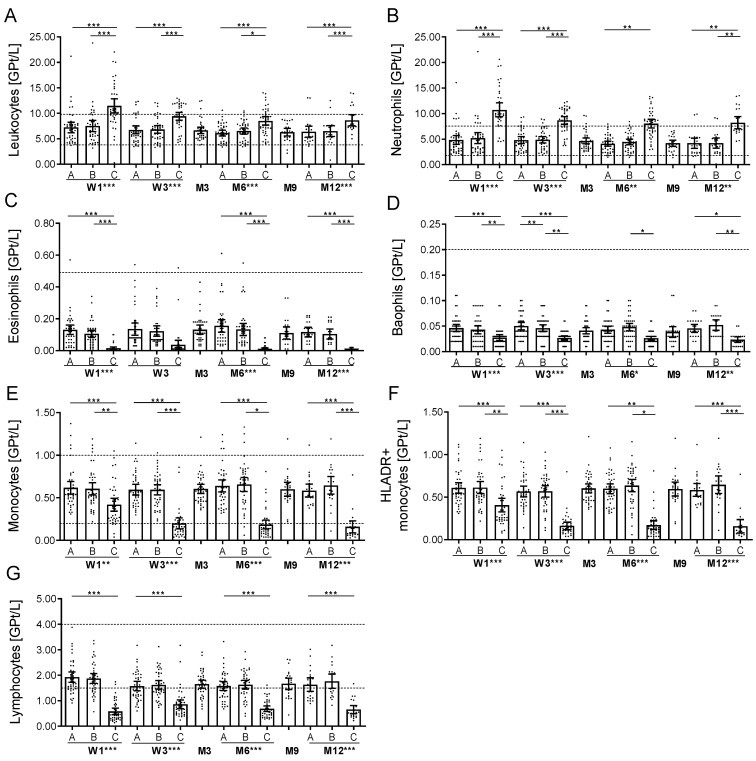
Routine blood analysis during OCR infusions. Cell count of leukocytes (**A**), neutrophils (**B**), eosinophils (**C**), basophils (**D**), monocytes (**E**), activated HLADR + monocytes (**F**) and lymphocytes (**G**) are presented. Blood samples were taken before infusions started [A], after MP [B] and after OCR [C] on each at the first two OCR infusions at week 1 (W1) and week 3 (W3), third infusion at month 6 (M6) and fourth infusion at month 12 (M12). Additional blood controls were done at month 3 (M3) and month 9 (M9). Bar/scatter plots with mean with 95%CI. Dotted line indicates the reference range. Asterisks indicate a statistically significant difference between selected time points and global effects (* *p* < 0.05, ** *p* < 0.01, *** *p* < 0.001).

**Figure 2 ijms-23-13759-f002:**
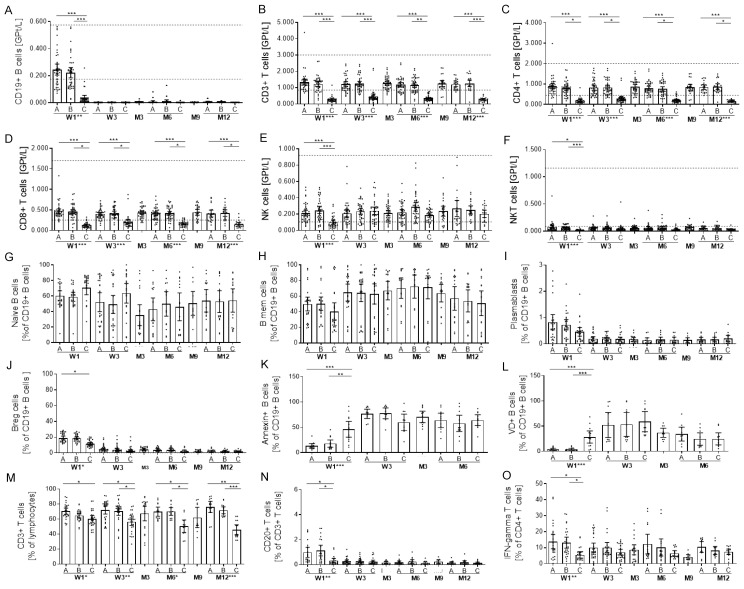
Cell count of selective immune cell sub-populations during OCR. (**A**–**F**) Absolute cell count of peripheral lymphocyte subsets evaluated by FACS analysis. Relative cell count of CD19 + CD27− naïve B cells (**G**), CD19 + CD27 + memory B cells (**H**), CD19 + CD27 + CD38 + CD138 + plasmablasts (**I**) and CD19 + CD27 + CD38 + CD5 + B regulatory cells (**J**) are depicted. (**K**,**L**) Apoptotic B cells (annexin, A) and death cells (viability dye, VD) were investigated in a selected subgroup (*n* = 10) and are presented. Relative count of CD3^+^ T cells (**M**), CD20 CD3+ T cells (**N**) and IFNgamma+ CD4+ T cells (**O**) are presented. Analysis was performed before infusions started [A], after MP [B] and after OCR [C] on each at the first two OCR infusions at week 1 (W1) and week 3 (W3), third infusion at month 6 (M6) and fourth infusion at month 12 (M12) and at month 3 (M3) and month 9 (M9) as well. Bar/scatter plot with mean with 95% CI are presented. Asterisks indicate a statistically significant difference between selected time points and global effects (* *p* < 0.05, ** *p* < 0.01, *** *p* < 0.001).

**Figure 3 ijms-23-13759-f003:**
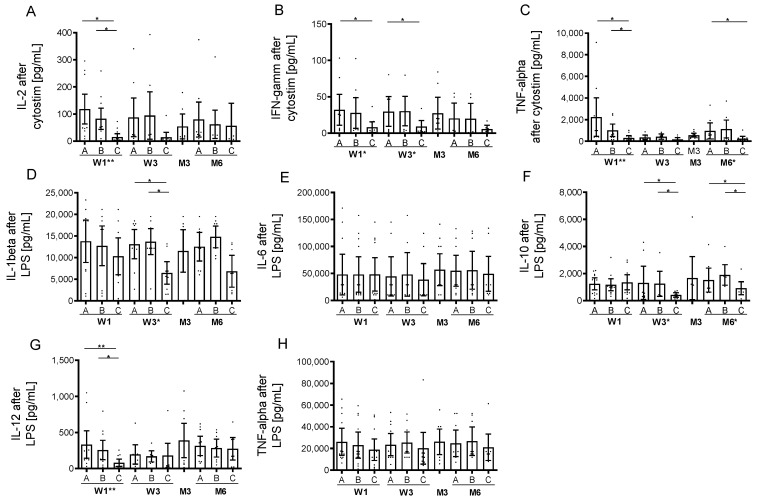
Cytokine release upon stimulation of cells during OCR treatment. Cytokine levels in cultured cell supernatants after ex vivo cytostim stimulation (**A**–**C**) and after ex vivo LPS stimulation (**D**–**H**) are depicted (*n* = 13). Analysis was performed at baseline [A], after MP [B], after OCR [C] at week 1 (W1) and week 3 (W3), month 3 (M3) and month 6 (M6). Mean values with 95% CI are presented. Asterisks indicate a statistically significant difference between selected time points and global effects (* *p* < 0.05, ** *p* < 0.01).

**Table 1 ijms-23-13759-t001:** Vital signs during OCR infusions.

	T0	T1	T2	T3	T4	T5
W1						
BPsyst/	127.6 (16.2)/	120.2 (22.2)/	123.4 (13.7)/	124.0 (14.0)/	125.6 (13.0)/	125.5 (13.4)/
BPdiast	82.6 (10.4)	80.3 (9.6)	78.3 (10.5)	79.2 (10.4)	79.1 (11.2)	78.2 (9.2)
[mmHg]						
HR [bpm]	78.3 (11.3)	77.8 (14.3)	76.7 (11.9)	81.0 (13.8)	84.1 (14.6)	84.7 (14.5)
BT [°C]	36.4 (0.5)	36.5 (0.4)	36.6 (0.3)	36.7 (0.4)	36.7 (0.4)	36.6 (0.4)
W3						
BPsyst/	127.6 (15.2)/	120.7 (15.1)/	121.6 (14.7)/	123.7 (14.7)/	125.4 (14.6)/	120.8 (11.6)/
BPdiast	81.9 (10.7)	78.5 (11.1)	78.4 (10.7)	79.2 (11.3)	77.6 (11.0)	77.3 (13.6)
[mmHg]						
HR [bpm]	78.1 (9.7)	73.9 (10.2)	72.6 (8.6)	75.0 (11.5)	79.1 (11.5)	87.1 (11.1)
BT [°C]	36.4 (0.3)	36.5 (0.4)	36.4 (0.4)	36.5 (0.4)	36.4 (0.3)	36.5 (0.4)
W6						
BPsyst/	125.4 (12.0)/	122.0 (13.9)/	122.5 (15.5)/	122.8 (15.5)/	120.0 (14.6)/	126.0 (15.7)/
BPdiast	81.3 (8.9)	80.2 (9.9)	77.6 (9.4)	77.9 (9.7)	76.7 (10.0)	78.6 (10.1)
[mmHg]						
HR [bpm]	78.4 (10.0)	71.9 (10.1)	72.7 (10.7)	75.1 (10.7)	76.5 (9.4)	80.7 (11.8)
BT [°C]	36.5 (0.3)	36.5 (0.3)	36.6 (0.3)	36.6 (0.3)	36.5 (0.4)	36.6 (0.3)
W12						
BPsyst/	125.2 (14.0)/	126.3 (11.6)/	122.8 (10.4)/	125.5 (11.3)/	124.4 (10.5)/	125.0 (10.3)/
BPdiast	84.7 (10.1)	85.0 (9.0)	80.8 (7.7)	82.6 (6.2)	81.6 (7.5)	82.0 (5.7)
[mmHg]						
HR [bpm]	76.6 (8.5)	73.9 (8.4)	74.6 (8.5)	76.7 (7.3)	77.5 (8.3)	81.5 (7.9)
BT [°C]	36.4 (0.2)	36.5 (0.2)	36.5 (0.2)	36.5 (0.2)	36.5 (0.3)	36.4 (0.2)

Vital parameters were evaluated before (T0), after MP (T1), each hour of OCR infusion (T2–T4) and at one hour follow up (T5). Data from the first two OCR infusions at week 1 (W1) and week 3 (W3), third infusion at month 6 (M6) and fourth infusion at month 12 (M12) are presented. Mean ± standard deviation. BP, blood pressure; bpm, beat per minute; BT, body temperature; HR, heart rate.

**Table 2 ijms-23-13759-t002:** Serum cytokine levels during OCR infusion.

	W1			W2			M3	M6			M9	M12		
	A	B	C	A	B	C		A	B	C		A	B	C
IL-1beta	11.2 (19.1)	10.4 (19.0)	11.1 (18.9)	12.3 (19.2)	10.1 (19.1)	9.3 (17.6)	7.1 (14.2)	3.8 (2.5)	3.3 (1.5)	3.8 (2.8)	3.2 (1.6)	3.2 (1.5)	3.5 (2.3)	3.4 (1.8)
IL-2	2.5 (2.8)	2.3 (2.8)	2.4 (2.6)	4.4 (7.0)	2.5 (2.7)	2.5 (2.5)	1.7 (1.4)	1.4 (0.8)	1.3 (0.5)	1.7 (1.2)	1.5 (0.9)	1.7 (1.3)	1.2 (0.6)	1.5 (0.7)
IL-4	7.5 (8.7)	6.9 (8.3)	5.6 (8.2)	7.5 (7.7)	7.3 (8.7)	5.1 (7.0)	4.5 (6.1)	4.1 (5.7)	3.8 (4.7)	3.5 (1.7)	3.1 (4.5)	3.7 (4.7)	3.7 (4.3)	3.4 (1.3)
IL-5	1.1 (0)	1.1 (0)	1.1 (0)	1.1 (0)	1.1 (0)	1.1 (0)	1.1 (0)	1.1 (0)	1.1 (0)	1.1 (0)	1.1 (0)	1.1 (0)	1.1 (0)	1.1 (0)
IL-6	5.8 (12.0)	7.8 (14.1)	16.9 (20.9)	5.1 (12.1)	5.2 (12.2)	4.7 (10.9)	3.3 (3.6)	2.7 (2.2)	3.0 (2.5)	2.9 (2.5)	3.4 (3.6)	3.6 (4.1)	3.7 (3.9)	3.4 (3.8)
IL-9	4.4 (6.2)	4.4 (6.6)	5.5 (6.5)	6.2 (7.7)	4.4 (5.7)	5.0 (5.4)	2.5 (3.0)	2.8 (3.7)	2.5 (2.9)	2.8 (3.5)	2.0 (1.4)	3.6 (4.7)	3.1 (4.3)	3.1 (3.3)
IL-10	6.8 (5.9)	6.2 (5.2)	128.0 (132.9)	8.5 (8.5)	5.4 (4.3)	21.5 (32.1)	5.5 (3.9)	3.4 (2.3)	3.8 (2.0)	28.9 (21.9)	1.9 (2.2)	5.8 (5.1)	3.4 (2.1)	23.6 (19.0)
IL-12	38.65 (54.6)	37.3 (57.6)	30.6 (61.3)	44.3 (66.1)	35.5 (58.9)	24.8 (54.5)	21.7 (33.4)	13.4 (21.5)	11.8 (17.3)	8.6 (8.6)	11.3 (20.1)	13.3 (18.1)	11.6 (17.6)	7.5 (9.0)
IL-13	3.7 (5.1)	3.5 (5.0)	3.4 (5.4)	5.0 (6.6)	3.6 (5.0)	3.8 (4.9)	2.4 (3.6)	1.7 (0.9)	1.5 (0.8)	1.7 (1.1)	1.5 (0.7)	1.9 (1.1)	1.7 (1.0)	1.8 (0.9)
IL-17	5.4 (6.1)	5.2 (6.2)	3.4 (3.3)	8.5 (8.6)	5.6 (5.3)	3.6 (3.2)	3.0 (2.5)	2.9 (4.5)	2.5 (3.6)	1.6 (1.1)	2,.4 (3.3)	2.7 (3.6)	2. 8 (3.5)	1.7 (1.1)
IL-22	4.3 (6.9)	4.3 (7.3)	5.1 (7.3)	5.1 (7.1)	4.6 (6.6)	5.0 (7.4)	4.2 (7.2)	3.1 (3.5)	2.4 (2.0)	2.5 (2.3)	2.5 (2.7)	2.5 (2.1)	2.3 (1.7)	2.1 (1.4)
IFN	5.1 (7.0)	4.4 (5.7)	8.1 (4.4)	5.9 (8.7)	3.8 (5.1)	3.6 (4.8)	2.6 (4.1)	2.1 (2.9)	1.7 (1.3)	2.1 (2.0)	2.1 (2.5)	1.9 (1.6)	1.7 (1.4)	1.8 (1.4)
TNF	7.4 (2.9)	9.7 (10.8)	24.3 (9.9)	6.6 (2.9)	6.0 (2.4)	5.3 (1.7)	6.5 (3.0)	5.8 (2.8)	5.4 (1.8)	6.3 (3.6)	6.9 (3.9)	6.0 (1.7)	5.5 (1.0)	5.8 (1.3)

Presented cytokine levels are mean +/− SD (standard deviation); significant changes are highlighted in grey (*p* < 0.05). Patients below the predefined measurable cutoff were set to the half of this value. Predefined cutoff values reflect the laboratory manufacturer’s instructions (pg/mL): IL-1beta: <4.6; IL-2: <2.02; IL-4: <2.87; IL-5: <2.17; IL-6: 2.0; IL-9: <2.05; IL-10: <1.92; IL-12: <1.80; IL-13: <2.04; IL-17A: <1.97; IL-22: <2.07; IFNgamma: <2.07; TNFalpha: 5.0. Baseline [A], after MP [B], after OCR [C] at infusions week 1 (W1), week 3 (W3), month 6 (M6) and month 12 (M12) and at month 3 (M3) and month 9 (M9).

**Table 3 ijms-23-13759-t003:** Patient characteristics.

Age, years (mean, SD)	41.6 (10.3)
Female (no, %)	28 (62.2)
Disease duration, years (mean, SD)	6.6 (7.1)
Disease course (no, (%))	
RMS	34 (76.0)
PMS	11 (24.0)
EDSS (mean, SD)	
RMS	3.1 (1.8)
PMS	4.5 (1.7)
Pre-treatment (no, %)	
none	17 (37.8)
injectables	10 (22.2)
fingolimod	6 (13.3)
dimethylfumarate	3 (6.7)
teriflunomide	3 (6.7)
natalizumab	2 (4.4)
daclizumab	4 (8.9)

Injectables (glatiramer acetate, interferon-beta).

## Data Availability

Data not provided in the article may be shared at the request of any qualified investigator.

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
