# Peer review of "Acute Effects of Ocrelizumab Infusion in Multiple Sclerosis Patients"

_ijms, 2022, doi:10.3390/ijms232213759_

Round 1

Reviewer 1 Report

Akgün and colleagues reported on immunological changes following treatment with humanized anti-CD20 antibody in multiple sclerosis. The topic is not particularly interesting nor novel. Not least, authors did not seem to know (or simply decided not to reference to) previous studies evaluating short and long term immunological effects of ocrelizumab, coming from both clinical trial extensions and the real world. That said, I have also tried more in-depth revision of the manuscript, but a number of oversights suggest the authors have put little effort in the manuscript. It cannot be expected that the effort to perform such revisions is up to the reviewers. For instance:

-          The very first sentence of the abstract states that “B cell depleting therapies as ocrelizumab (OCR) are highly effective in multiple sclerosis (MS) presenting with infusion-related reactions (IRR) especially in the treatment start”. This sentence is not clear to me, and I guess something is missing. Why ocrelizumab should especially effecting in MS patients with IRR?

-          The very first part of the introduction (first eight lines) are made of the standard format instructions, and should be removed.

-          Authors claims they have measured plasma cells in the peripheral blood. However, plasma cells are not meant to be present in the peripheral blood, and I am not sure what the authors have actually measured and how, especially in the absence of a disclosed gating strategy.

I will be happy to have a further look at the manuscript if authors are willing to fully acknowledge previous results from clinical trial extension and real-world studies (with larger sample and longer follow-up), subsequently identifying what gaps they are actually filling in. Then, some additional effort should be put throughout the manuscript to remove oversights, showing they actually care about it.

Author Response

Akgün and colleagues reported on immunological changes following treatment with humanized anti-CD20 antibody in multiple sclerosis. The topic is not particularly interesting nor novel. Not least, authors did not seem to know (or simply decided not to reference to) previous studies evaluating short and long term immunological effects of ocrelizumab, coming from both clinical trial extensions and the real world. That said, I have also tried more in-depth revision of the manuscript, but a number of oversights suggest the authors have put little effort in the manuscript. It cannot be expected that the effort to perform such revisions is up to the reviewers. For instance:

Comment: The very first sentence of the abstract states that “B cell depleting therapies as ocrelizumab (OCR) are highly effective in multiple sclerosis (MS) presenting with infusion-related reactions (IRR) especially in the treatment start”. This sentence is not clear to me, and I guess something is missing. Why ocrelizumab should especially effecting in MS patients with IRR?

Response: The sentence was restructured to make clear that this treatment is associated with high efficacy and side effects as IRR.

Comment: The very first part of the introduction (first eight lines) are made of the standard format instructions, and should be removed.

Response: Thank you for that remark. We apologize for that mistake and corrected that in the current version.

Comment: Authors claims they have measured plasma cells in the peripheral blood. However, plasma cells are not meant to be present in the peripheral blood, and I am not sure what the authors have actually measured and how, especially in the absence of a disclosed gating strategy.

Response: Gating strategy is defined in the figure legend of the presented graph. The presence of peripheral plasma cells is contradictory discussed in the literature as small number of plasma cells can also be found in peripheral blood especially in clinical conditions including immune disorders (Caraux A. et al. Circulating human B and plasma cells. Age-associated changes in counts and detailed characterization of circulating normal CD138− and CD138+ plasma cells. Haematologica Vol. 95 No. 6 (2010): June, 2010; A. Horst et al. Detection and characterization of plasma cells in peripheral blood: correlation of IgE+ plasma cell frequency with IgE serum titre. Clin Exp Immunol. 2002 Dec;130(3):370-8). As we want to focus on acute effects of ocrelizumab infusion and not B cell depleting mechanism itself in our study, we decided to shorten and rework that paragraph and figure.

Comment: I will be happy to have a further look at the manuscript if authors are willing to fully acknowledge previous results from clinical trial extension and real-world studies (with larger sample and longer follow-up), subsequently identifying what gaps they are actually filling in. Then, some additional effort should be put throughout the manuscript to remove oversights, showing they actually care about it.

Response: Thank you for the review of our manuscript. In our study, we are not dealing with B cell depleting effects in the long-term as many other studies have done so, but want to discuss acute effects of ocrelizumab infusions to understand how to deal with side effects and premedication including infusion management. That is in the analogy of our approach to acute effects of alemtzuzmab (Thomas, K., Eisele, J., Rodriguez-Leal, F. A., Hainke, U. & Ziemssen, T. Acute effects of alemtuzumab infusion in patients with active relapsing-remitting MS. Neurology Neuroimmunol Neuroinflammation 3, NA; (2016). Though the main part of our manuscript is dealing with these acute effects and only (to a smaller extent) long-term data in the trial and real-world cohorts. In addition, our manuscript was reworked and oversights corrected. However, it is impossible for us to fully address your response as comments do not fit to the main focus of our manuscript.

Reviewer 2 Report

This to is a solid desription of immune responses to ocrelizumab treatment in group of MS patients. The major findings are that cytokine secretion is limited mainly to the first administration of ocrelizumab and that populations other than the target CD20 B cells are minimally affected. The discussion does detail some of this issue. 

Major points: Ocrelizumab has been in clinical use for more than 12 years and the major findings regarding the adverse reaction to the initial infusion and selctivity to B cells have essentially been described already in previous publications such as reference 1 in this manuscript. The more detailed and methodical analysis offered by the present study adds certainly more detail and in depth analysis and thus does deserve publication. 

Figures 2, 3 and 4 are very croweded and with very small fonts and are barely legible in their present form. They must be modified since it is almost impossible to understand the data presented without enlarging the figures on screen. 

Minor points: 

The first lines of the introduction 24-32 are the instructions probably found in the template of the submission form and should of course be ommitted. 

Author Response

This to is a solid desription of immune responses to ocrelizumab treatment in group of MS patients. The major findings are that cytokine secretion is limited mainly to the first administration of ocrelizumab and that populations other than the target CD20 B cells are minimally affected. The discussion does detail some of this issue. 

Comment: Major points: Ocrelizumab has been in clinical use for more than 12 years and the major findings regarding the adverse reaction to the initial infusion and selectivity to B cells have essentially been described already in previous publications such as reference 1 in this manuscript. The more detailed and methodical analysis offered by the present study adds certainly more detail and in depth analysis and thus does deserve publication. Figures 2, 3 and 4 are very crowded and with very small fonts and are barely legible in their present form. They must be modified since it is almost impossible to understand the data presented without enlarging the figures on screen. 

Response: Thank you very much the response. We reworked the structure of the figures 2-4 for better understanding and shortened figure 4 to present more precise and relevant data.

Comment: Minor points: The first lines of the introduction 24-32 are the instructions probably found in the template of the submission form and should of course be ommitted. 

Response: Thank you very much for that remark. We apologize for that mistake and corrected that in the current version.

Round 2

Reviewer 1 Report

Authors are unwilling to address my previous concerns.

Author Response

Reviewer/Editor comments:

Comment 1: The aim of the study was to assess acute infusion related side effects of ocrelizumab in MS patients and observe the patients up to 12 months (6 infusions). The aim is not well laid out and the authors present a large amount of negative routine laboratory tests so that the message gets lost among a descriptive evaluation/presentation which could be presented in a more concise way. 
The authors need to rework the aim according to SMART (Specific, Measurable, Attainable, Relevant and Timely) criteria.

The data could reveal far more information and actually the authors claim that their aim was indeed to answer further questions so see how a IRR (infusion related reaction) impacted on the outcome of subsequent infusion and outcome of ocralizumab mediated B-cell depletion. However, this aim was not met by the descriptive analysis of the data which is at a very basic level (some examples how to revise the data analysis and presentation are given below).
Throughout the paper individual data points of the patients (n =45) are not presented, and the figures do not reveal the time courses of individual patients which would be crucial to assess the predictive value of the IR response to the 1st-dose.

In sofar, I agree with Reviewer #1 that the paper is not particularly "well written"/ "well-analyzed". The message is also not novel. IRR effects of ocrelizumab are well described. The add-on information of the present manuscript is basically the higher testing frequency of multiple parameters/readouts, which are so far described but not analysed to reveal novel insight
The major criticism of Reviwer#1, however, (the authors should elaborate on long term ocrelizumab effects) does not strike, because this was not the aim of the present study. The authors have addressed the other minor comments. 

Response 1: The aim of that study was to discuss and present the infusion-associated acute reactions in ocrelizumab treated patients in relation to clinical signs and management.

IRR are a known phenomena in cell depleting therapies especially when using monoclonal antibody treatment. Different reports present the prevalence and management of IRR including the initial clinical trials. However, in clinical practice and after approval further questions appear as patient characteristics in real world settings differ from patients included in the study trial.

The most prominent example is the post-marketing experience with alemtuzumab, a monoclonal antibody against CD52 depleting T and B cells. Within clinical trials frequency of IRRs and management of IRRs were discussed. But only post-marketing studies demonstrated how fast T and B cells were depleted and that depletion of these cell subsets was accompanied by a cytokine release that affected conventional lab testing including chemical chemistry and lead to misinterpretation of lab parameters in clinical practice in the period early after treatment start. In addition cases with severe infections post-infusion appeared after approval. New data on acute infusion-associated effects on peripheral immune cell subsets demonstrated that patients are relevantly immunosuppressed early after infusion which is reconstituted after one months and though was not realized in the initial trials having looked on broader time intervals.

Since approval of ocrelizumab and in the real world cohort further questions appear: why do selected patients present severe infections early after infusions start and how can we manage that, how intense is the IRR cytokine release and may this impact interpretation of lab parameters within first days after infusion, can we treat patients without corticosteroid premedication in case there are contraindications in corticosteroid use? These aspects are of so much importance in clinical use and everyday clinical practice as we used to treat an older cohort with relevant comorbidities and other immunosuppressive pretreatment constellation compared to patients in clinical trials.

That is why we have chosen a descriptive-explorative approach in our study. Although IRR are a known phenomenon during ocrelizumab treatment, we wanted to address especially these clinical questions. We evaluated 4 ocrelizumab infusions within a one year period to detect if there are differences in the longitudinal follow up of these infusion-associated effects during first versus follow up infusions. Therefore we present multiple parameters to highlight the complexity of this topic and to address each aspect clinicians are to be faced with in clinical practice. However it was not our aim to correlate IRR with the clinical outcome (e.g. disease activity) or intensity of B cell depletion.

We added details on that in the manuscript to be more clear regarding the rational and aim of our study.

Presentation of the data:

Comment 2: Figure 1 One would like to see individual data for example in multiple line plots in a 4-column graph for 1W, 3W, 6M, 12M, and three rows for each parameter. Alternatively, the data could be shown in a table. 

Response 2: Thank you for your suggestion. In Figure 1 we present the variation of vital parameters (blood pressure, heart rate, body temperature) at each infusion time point (W1, W3, M6, M12). Based on the standardized infusion and management protocol, these parameters will be checked hourly during the 6 hours infusion procedure. To document every parameter, at every time point for each patient in individual line blots one graph would be to crowded. As mentioned in the manuscript, no relevant change could be proved in our cohort during the infusions. As the last version of figure 1 was not clear enough for the reader, we have changed it into a table as suggested by the editor.

Comment 3: Figure 2: One would like to see individual data. There are 45 patients, each with 14 time points. It is important to see how a specific patient responds to the 1st and subsequent infusions, which is not revealed by the present presentation. The time courses could then be clustered e.g. with K-means 3-4 clusters to find out if patients with similar time courses have something in common. One could then address if patients with strong 1st-dose IRR have a good/bad/normal outcome at one year in terms of overall side effects and B-cell depletion efficacy

Figure 4 like in figure 2 one would like to see a cluster analyses of time courses and parameters and multivariate analyses to reduce the dimensionality and assess which parameters have the highest impact and separate patients with/without 1st-doese IRR. 
Figure 3/4: again individual data points should be revealed e.g. as box/scatter plots. 

Response 3: Our analyes/data are in fact very descriptive in nature. In figures, descriptions of the raw data was done using Box-whisker-plots with 5-95 percentiles. In addition, p-values were calculated via GLMM (generalized linear mixed (regression) models) and incorporated into the figures. It is worth noting that using GLMMs is a very contemporary, but also complex method for statistical evaluations which can handle e.g. extreme values, skewed distributions of the outcomes, intra-individual correlations (longitudinal data) and to some extent missing values.

Multiple regression analyses were done using GLMMs with a basically univariate approach (time as independent variable). Thereby, we considered each outcome separately, which means we did not consider pairwise correlations between the outcomes. However, we believe that this approach is in line with our research question (see response 1).

Nevertheless, to gain a basic understanding of the interdependence of the outcomes of interest, we calculated Spearman’s correlation coefficients – both longitudinally (between A to C) and cross-sectionally. To further address the editor´s further comments, we changed the figures to bar/scatter plots in order to present individual data points. In addition, in the current version of the manuscript, we considered another multivariable analysis that extended the previous GLMM by including not only the exploratory variable ‘time’ (A, B, C), but also the ‘IRR group’ (yes, no) and the ‘interaction’ of both (IRR x time). Again, it was not our aim to link IRR with outcomes at one year with focus on side events and B cell depletion. This is not that relevant from a clinical perspective as stated above. Though we do not think it is useful to add such an evaluation in our manuscript.

Comment 4: Figure 2/3: there is one outlier with high levels at baseline. It if were the same patient, it would be recognized as "outlier" by statistical tests and should be taken out from the figures to improve the presentation/visibility of the other patients. The outlier could be explained in the figure legend

Response 4: The graphs present Box-whisker-plots with 5-95 percentiles as stated in the figure legend. Based on the definition of the chosen graph, the dots do not present an outlier but data points above the 95th or below the 5th percentile. Here we present biomarkers in a real-world patient cohort. Though its not unusual that single parameters are above or below the defined reference range. Thus, they are extreme values, but not necessarily outliers. Furthermore, outcomes are often skewed (e.g right-skewed) which must be taken into account when considering the probabilities for extreme values/outliers. In our evaluation, only (lab-)technical outliers were excluded. We feel that exclusion of further values does interfere with the natural distribution of biomarkers in the lab-testing and may lead to misinterpretation of our real-word data. As stated above, figures were restructured to be more clear regard the results.

Comment 5: Figure 3 shows routine lab tests, all without change and could be moved to a supplement

Response 5: As stated above, the aspect if routine lab parameters are affected by cell depleting therapies is very relevant for physicians in clinical practice. In contrast to other cell depleting therapies also used in multiple sclerosis therapy, here we could proof that routine lab testing and interpretation of these values is not changed by ocrelizumab. However we understand that routine data do not present relevant changes, so figure 3 is moved to supplement as suggested.  

Comment 6: Figure 4: looks like FACS data (not explained in the figure legend), the gating strategy needs to be demonstrated e.g. with exemplary scatter plots

Response 6: Yes, figure 4 presents FACS data, which is clearly stated in the methods section, gated markers for each cell are presented in the figure legend. For additional transparency, gating strategy was added as supplement figure.

Comment 7: Figure 5: again individual data points shall be revealed. The layout of the figure is not well chosen (better for example four columns/three rows). The labels are tiny, the subpanels need headlines and the side-by-side readouts should be matched. In so far, the criticism of Reviewer #2 (not readable/unclear figures) is not resolved. 

Response 7: Figure 5 was restructured.

Comment 8: Table 1: legend does not explain the method for cytokine analysis and the thresholds for "increase" (+, ++). What about decreases?

Response 8: The authors understand that the old version of the table was not completely clear. We reworked the table completely now presenting distinct values to be more clear regard the results.

Comment 9: bar charts need to be replaced by bar/scatter, box/scatter or scatter throughout to reveal sample sizes, individual levels and biological variability

Response 9: Figures were replaced and restructured.

Comment 10: Presently, the readouts are not associated with each other. For example, the present presentation/analysis does not reveal if leukocyte numbers are associated with high cytokine levels, or are independent parameters which could be addressed for example with multiple regression analyses or multivariate analyses (such as PLS-DA, OPLS, PCA....) 

Response 10: See response #3. Results are included in the current version of the manuscript.

Comment 11: Writing: in part complicated and in part unclear sentences. For example last sentence Abstract. The present study does not show a comparison with alternative treatments. Hence the last sentence of the Abstract is misleading. 
" In comparison to other depleting intravenous therapies, our data demonstrate only mild changes at first OCR infusion which are not present any more during long-term treatment"
The conclusion should focus on the present study: for example Our data demonstrate that the 1st dose infusion of OCR causes IRR in subsets of patients but does not recur with subsequent treatments. 
The discussion should be shortened to focus on the presented results. 
Conclusion discussion: it is not a "mechanistic" real world study. Indeed the study is descriptive in nature and so far does not reveal which of the blood tests /cells accounted for IRR symptomatology. The authors even write that IRR positive patients had not particularly high cytokines.

Response 11: Thank you very much for that advise. The manuscript text was shortened, rewritten and misleading parts corrected. We hope to be now clearer regarding the aim and the rational of that manuscript